# A Review on Risk Management of Coronavirus Disease 19 (COVID-19) Infection in Dental Practice: Focus on Prosthodontics and All-Ceramic Materials

Roberto Sorrentino [1] , Michele Basilicata [2], Gennaro Ruggiero [1] , Maria Irene Di Mauro [1], Renato Leone [1,*] , Patrizio Bollero [3] and Fernando Zarone [1]

1 Department of Neurosciences, Reproductive and Odontostomatological Sciences, Division of Prosthodontics and Digital Dentistry, University "Federico II" of Naples, 80131 Naples, Italy; roberto.sorrentino@unina.it (R.S.); gennaro.ruggiero2@unina.it (G.R.); mariadimauro94@gmail.com (M.I.D.M.); zarone@unina.it (F.Z.)
2 Department of Experimental Medicine and Surgery, University of Rome "Tor Vergata", 00133 Rome, Italy; michele.basilicata@ptvonline.it
3 Department of Systems Medicine, University of Rome "Tor Vergata", 00133 Rome, Italy; patrizio.bollero@ptvonline.it
* Correspondence: renato.leone@unina.it; Tel.: +39-081-7463018

**Abstract:** Background: A novel β-coronavirus infection (COVID-19) was first detected in Wuhan city, spreading rapidly to other countries and leading to a pandemic. Dental professionals and patients are exposed to a high risk of COVID-19 infection, particularly in the prosthodontic practice, because of the bio-aerosol produced during teeth preparation with dental handpieces and the strict contact with oral fluids during impression making. This paper aimed to provide an overview to limit the risk of transmission of COVID-19 infections during prosthetic procedures in dental offices. Methods: An electronic search was conducted on the electronic databases of PubMed/Medline, Google Scholar, Embase, Scopus, Dynamed, and Open Grey with the following queries: (COVID-19) AND/OR (SARS-CoV-2) AND/OR (Coronavirus) AND/OR (contaminated surface) AND/OR (cross-infection) AND/OR (Prosthodontics) AND/OR (dental ceramic) AND/OR (glass-ceramic). A manual search was performed as well. Results: From the 1023 collected records, 32 papers were included. Conclusions: Dental offices are at high risk of spreading SARS-CoV-2 infection due to the close contact with patients and continuous exposure to saliva during dental procedures. Therefore, pre-check triages via telephone, decontamination, the disinfection of impressions, the sterilization of scanner tips, and the use of specific personal protective equipment, dental high-speed handpieces with dedicated anti-retraction valves, and effective mouthwashes are strongly recommended.

**Keywords:** coronavirus; COVID-19; SARS-CoV-2; prosthodontics; airborne; dental impression; pandemic; prevention



## 1. Introduction

In December 2019, severe acute respiratory syndrome coronavirus 2 (SARS-CoV-2), was first reported in Wuhan, Hubei province (China), and rapidly spread over 24 countries, leading the World Health Organization (WHO) to declare this severe pneumonia a global emergency on 30 January 2020 [1]. From 31 December 2019 to 17 June 2022, over 535,863,000 confirmed cases of coronavirus infectious disease-19 (COVID-19) have been reported, including approximately 6,315,000 deaths, and these numbers are increasing daily (Figures 1 and 2) [2].

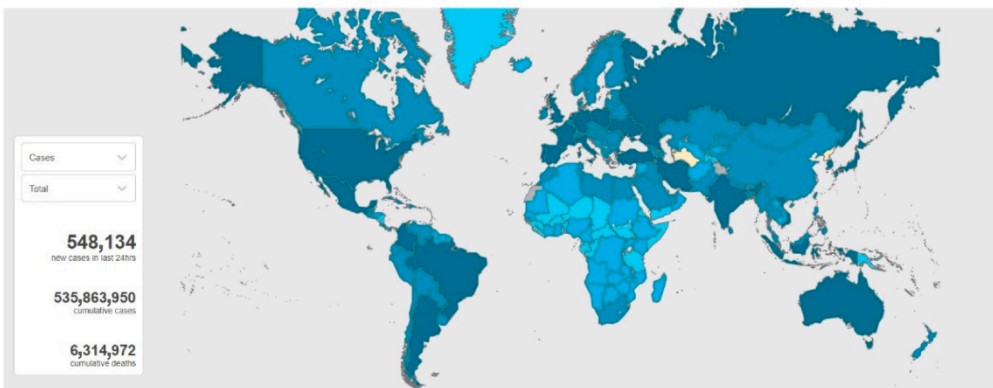

Globally, as of 5:24pm CEST, 17 June 2022, there have been 535,863,950 confirmed cases of COVID-19, including 6,314,972 deaths, reported to WHO. As of 16 June 2022, a total of 11,902,271,619 vaccine doses have

**Figure 1.** A screenshot of the interactive dashboard of COVID-19 global cases by the World Health Organization. This dashboard is continually updated and can be accessed at https://covid19.who.int/ (accessed on 17 June 2022).

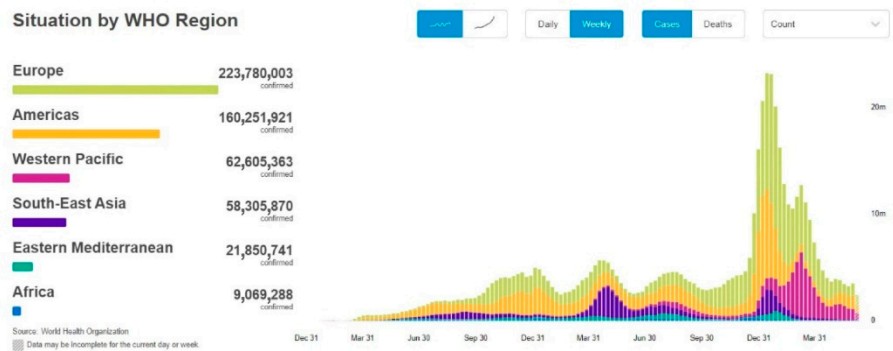

**Figure 2.** A screenshot of the COVID-19 situation by WHO Region. This dashboard is continually updated and can be accessed at https://covid19.who.int/ (accessed on 17 June 2022).

SARS-CoV-2, of the family Coronaviridae, is a single-stranded RNA beta coronavirus that is characterized by a diameter of 50–200 nm and probably originated from bats and pangolins [3]. It shares 85–92% nucleotide sequence homology with the pangolin coronavirus (CoV) genome and 96.2% nucleotide homology with bat CoV RaTG13, confirming the zoonotic origin of the virus [4]. The animal-to-human transmission event, the spillover phenomenon of SARS-CoV-2, can be plausibly imputed to the sale and killing of wildlife species at the Huanan Seafood Wholesale Market, where the initial cases of COVID-19 emerged [4]. The comprehension of the initial dynamics of the infection and the identification of the animal source of SARS-CoV-2 would help to prevent future new zoonosis by strengthening the control of food and hygiene within live animal markets.

Since the first COVID-19 infection, some variants of SARS-CoV-2 have been found [5]. These variants are adaptive mutations in the viral genome that can alter the virus's pathogenic potential, and some of them were classified as variants of concern (VOCs) due to their public health implications [5,6]. In particular, VOCs have been associated with increased virulence or transmissibility, decreased neutralization by antibodies acquired through vaccination or natural infection, the ability to elude detection, and a reduction in therapeutic or vaccine efficiency [6]. Five SARS-CoV-2 VOCs have been recognized according to the WHO: Alpha (B.1.1.7, first report in the United Kingdom); Beta (B.1.351, first report in South Africa); Gamma (P.1, first report in Brazil); Delta (B.1.617.2, first report in India); and Omicron (B.1.1.529, first report in South Africa) [6]. Among the listed VOCs, the Omicron variant is the most severely altered, paving the path for increased transmissibility and partial resistance to COVID-19 vaccine-induced immunity (Figures 3 and 4) [5,7,8].

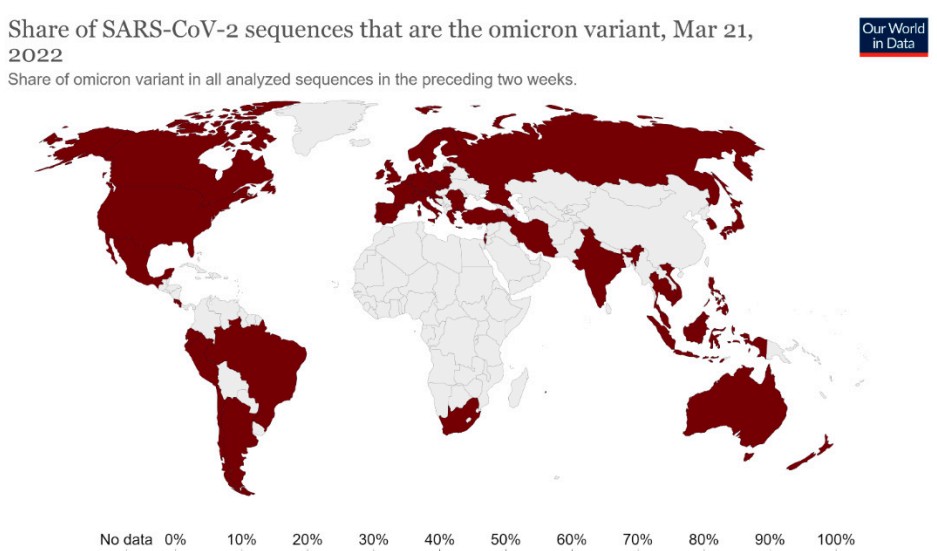

**Figure 3.** Share of SARS-CoV-2 sequences that are the omicron variant. This dashboard is continually updated and can be accessed at https://ourworldindata.org/grapher/covid-cases-omicron?country=GBR~FRA~BEL~DEU~ITA~ESP~USA~ZAF~BWA~AUS (accessed on 17 June 2022).

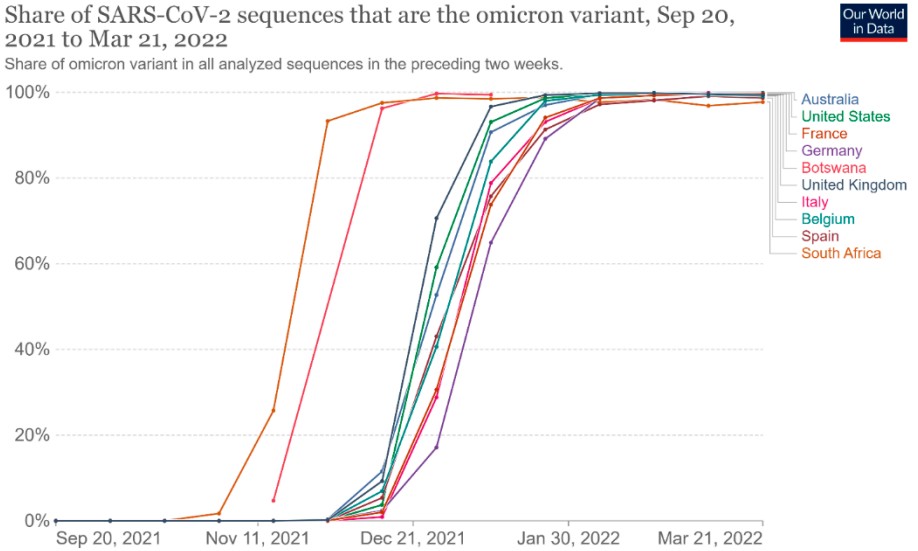

**Figure 4.** Share of omicron variant in all analyzed sequences. This dashboard is continually updated and can be accessed at https://ourworldindata.org/grapher/covid-cases-omicron?tab=chart&country=GBR~FRA~BEL~DEU~ITA~ESP~USA~ZAF~BWA~AUS (accessed on 17 June 2022).

The time from the infection to the onset of symptoms can vary from 2 to 14 days, and the median incubation period reported by the WHO was 5–6 days [9,10].

Older age and other risk factors, such as a history of underlying diseases and/or co-infections, play an important role in determining the severity of symptoms, leading to a higher risk to develop severe illness and death [11].

The clinical initial manifestations of COVID-19 can be aspecific. First of all, a large number of patients show symptoms of common cold, such as dry cough, sore throat, low-grade fever, or myalgia [11]. Less frequent manifestations of the infection may be nausea, diarrhea, dysgeusia, and persistent olfactory dysfunction (hyposmia) due to mucosal edema and nasal inflammation [12,13]. Some findings underlined the potential transmission of SARS-CoV-2 mediated by asymptomatic subjects, who may represent potential reservoirs for the spreading and re-emergence of the infection since the viral load in such patients was comparable to that of symptomatic patients [14].

Even if the mechanism of SARS-CoV-2 infection is not yet completely known, the high transmissibility of the virus can be partially explained by the higher affinity of the virus for cells located in the lower airways, where the virus binds using the host receptor for the conversion enzyme of angiotensin 2 (ACE2) and replicates, causing pneumonia [15].

The radiologic signs of bilateral pneumonia are abnormal ground-glass opacities found in chest X-ray and computed tomographic (CT) scans. The worst clinical picture patients can present is multiorgan dysfunction, acute renal failure, and acute respiratory distress syndrome (ARDS) [16].

People older than 60 years old and patients with chronic comorbidities (especially diabetes, high blood pressure, and cardiovascular diseases) were found to be more prone to develop severe clinical disease and fatal outcomes, which occurred only in critical cases. Fortunately, children often seem to show mild symptoms of COVID-19 [15].

SARS-CoV-2 has a human-to-human typical transmission by means of respiratory droplets and indirect contact, whereas the conjunctival and mother-to-fetus modes still need to be confirmed [17].

Some evidence suggested that infected individuals can spread infection through bodily secretions, such as the saliva and nasal fluid produced by talking, sneezing, and coughing. The high presence of the virus in saliva may be explained by the binding of SARS-CoV-2 with ACE2 receptors, which are highly concentrated in salivary glands [18–20].

Moreover, the contact of contaminated hands with the mucous membranes of the mouth and/or nose may lead to the onset of COVID-19 infection. The fecal–oral mode also may be another important route for nosocomial spread. Consequently, the disinfection of objects, handwashing, and social distancing (beyond six feet) are strongly recommended in order to control the community outbreak of the disease [17–20].

In particular, dental offices could be easily contaminated since the use of high-speed handpieces or ultrasonic instruments could cause the aerosolization of patients' secretions, such as saliva or blood. Thus, dental professionals are exposed to a high risk of contracting SARS-CoV-2 because social distancing is unachievable during dental procedures [21].

On the other hand, the increased susceptibility of patients in dental offices is another important concern: elderly age, diabetes, chemotherapy, pregnancy, or conditions of an impaired immune defense system may easily lead to a worsening clinical picture or fatal outcome in the case of SARS-CoV-2 infection [11,22].

Among the branches of dentistry, prosthodontics is the part of restorative dentistry concerned with the design, manufacture, and fitting of artificial replacements for missing teeth and the associated soft tissues. Many aspects and devices used during prosthetic procedures may offer the opportunity for cross-contamination, requiring careful attention and rigorous protocols for the prevention of infection spreading.

To date, there are scarce data in the scientific literature about the management of COVID-19 infection during prosthodontics procedures in order to prevent COVID-19 cross-infections.

The aim of the present review was to provide useful information to prevent the transmission of COVID-19 infections during daily prosthodontics practice in dental offices.

## 2. Methods

### 2.1. Search Strategy

The search was conducted in the following electronic databases: PubMed/Medline, Google Scholar, Scopus, Embase, Dynamed, and Open Grey. The authors of non-published or not electronically available articles were contacted by the authors of the present manuscript.

The following keywords and the Boolean operators AND/OR were used to pursue the research:

1. COVID-19;
2. SARS-CoV-2;
3. Coronavirus;
4. Contaminated surface;

5. Cross-infection;
6. Prosthodontics;
7. Dental ceramic;
8. Glass-ceramic.

One specific query was used for each electronic database in order to perform paper screening, as follows:

- Pubmed (Medline) = ("COVID-19" or "SARS-CoV-2" or "coronavirus" or "contaminated surface" or "cross-infection") and ("prosthodontics" or "dental ceramic" or "glass-ceramic");
- Google Scholar = title, abstract, keywords: "COVID-19" or "SARS-CoV-2" or "coronavirus" or "contaminated surface" or "cross-infection" and (prosthodontics or dental ceramic or glass-ceramic);
- Open Grey = (COVID-19) or (SARS-CoV-2) or (Coronavirus) or (contaminated surface) or (cross-infection) and (Prosthodontics or dental ceramic or glass-ceramic);
- Dynamed = (COVID-19; SARS-CoV-2; coronavirus) and/or (contaminated surface) and/or (cross-infection) and/or (prosthodontics or dental ceramic or glass-ceramic);
- Scopus = (TITLE-ABS-KEY ((COVID-19)) or TITLE-ABS-KEY ((SARS-CoV-2)) or TITLE-ABS-KEY ((coronavirus)) or TITLE-ABS-KEY ((contaminated and surface)) or TITLE-ABS-KEY ((cross-infection)) and TITLE-ABS-KEY ((prosthodontics or dental and ceramic or glass-ceramic)));
- Embase = ('COVID-19':ti,ab,kw or 'SARS-CoV-2':ti,ab,kw or coronavirus:ti,ab,kw or 'contaminated surface':ti,ab,kw or 'cross infection':ti,ab,kw) and (prosthodontics:ti,ab,kw or 'dental ceramic':ti,ab,kw or 'glass ceramic':ti,ab,kw).

In order to eliminate duplicates, the identified records' references were imported as a research information systems file into Mendeley (Mendeley Ltd., London, UK).

The "PRISMA 2009 Flow Diagram" was used for the screening process of the collected records (Figure 5) [23].

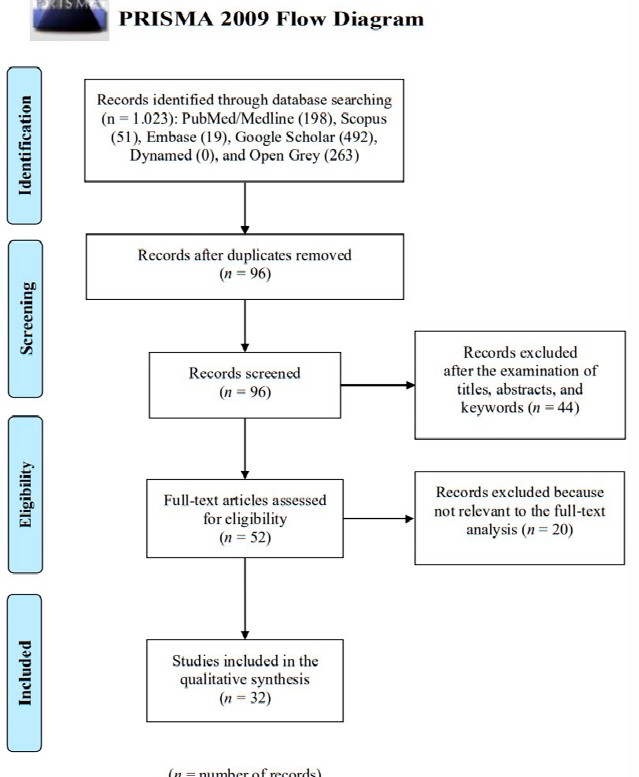

**Figure 5.** Prisma flow diagram for the records screening process.

## 2.2. Inclusion and Exclusion Criteria

The studies were considered eligible for the present review if they met the following inclusion criteria: (1) studies that provide information on how to limit the risk of transmission of COVID-19 infections during prosthetic procedures; (2) in vitro, in silico, or in vivo studies; (3) case reports; (4) systematic reviews or meta-analyses. Non-human animal studies were excluded. There were no restrictions on the date of publication or the language of the papers.

## 2.3. Data Extraction

To avoid a lack of relevant papers, the authors examined the reference lists of the identified records.

Three calibrated researchers (G.R., M.I.D.M., and R.L.) independently selected the studies, reading titles, abstracts, and keywords. The full text of each identified article was read to decide if it was suitable for inclusion. A majority criterion (i.e., two out of three) was used in case of disagreement among the investigators.

## 2.4. Calibration Process

Three reviewers (G.R., M.I.D.M., and R.L.), with the same level of experience, used a common random group of 30 references to conduct pilot calibration exercises on titles and abstracts using the inclusion and exclusion criteria. Subsequently, the researchers discussed which references were included and which ones were excluded. The reviewers aimed to obtain an agreement level of at least 80% for the papers. Before screening the entire list of acquired titles and abstracts, the process would have been repeated until they reached the required agreement level. In addition, the calibration method was performed on a random sample of 12 papers for a full-text screening of the included articles after reading titles and abstracts with the same agreement level.

## 3. Results

The literature search was concluded in December 2021, and the records included in the present paper were published between 1998 and 2021.

The electronic search produced 1023 records: 198 from PubMed/Medline, 492 from Google Scholar, 19 from Embase, 51 from Scopus, 0 from Dynamed, and 263 from Open Grey.

After removing all duplicates, the specified databases yielded 96 records. The reviewers eliminated 44 papers that did not match the inclusion criteria after examining the titles, abstracts, and keywords. The remaining 52 records were subjected to a full-text analysis. Then, 20 more were eliminated because they did not provide effective information on procedures to manage COVID-19 infection during prosthetic procedures. The remaining 32 papers were included in the present review.

The reviewers obtained levels of agreement of 90% on the title and abstract screening and 100% on full-text article screening after just one calibration session.

There was no disagreement among the search investigators about the records that were included.

Among the 32 included records, 25% were published in 2021, 31.25% in 2020, 18.75% between 2019 and 2010, and 25% between 2008 and 1998.

Neither randomized controlled trials (RCTs) nor long-term clinical studies were included in the present paper. The most interesting investigations were thoroughly analyzed by the reviewers and discussed below in order to draw some guiding principles based on the available published literature and the current knowledge on this topic.

## 4. Discussion

Despite the global efforts, the recent SARS-CoV-2 infection is difficult to contain due to its high level of contagiousness and its massive detection in saliva [19,24].

Several prosthetic procedures (i.e., impression making, interocclusal records, chairside prosthesis modifications) imply not only a strict closeness to patients but also handling many items contaminated with several microbial species.

### 4.1. Dental Impression

Dental impression and wax or silicone interocclusal records could be contaminated with patients' saliva and, just as frequently, blood (Figure 6).

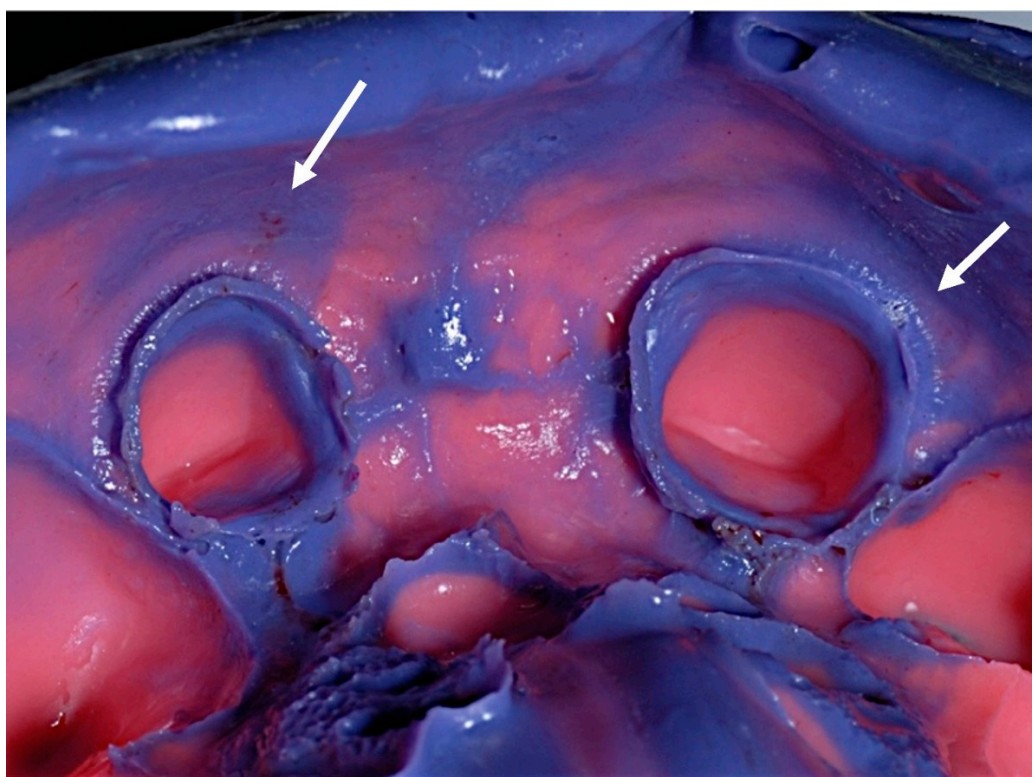

**Figure 6.** Debris of saliva and blood on a conventional elastomeric dental impression (arrows indicated).

Some investigations found low bacterial contamination (median number of $1.3 \times 10^2$ cfu/20 mm$^3$) on a high percentage of dental impressions (more than 70%), whereas higher contamination was found ($10^3$–$3.4 \times 10^4$ cfu/20 mm$^3$) on the resultant percentage of samples. Bacteria can be opportunistic or non-opportunistic species, and low-pathogenic species may promote the onset of latent infections and/or reactivate ones [25,26]. Regarding the viral contaminations of dental impressions, it is widely known that patients' body fluids (i.e., saliva and blood) might be potential reservoirs and sources of the disease, hence the urge to pursue meticulous protocols of decontamination and disinfection of the impressions before sending them to dental laboratories in order to reduce gypsum cast contamination with patient-derived microbes.

Regarding conventional dental impressions, a prior accurate debridement of saliva and blood is recommended by brushing and/or rinsing under running water to allow complete contact between the dental impressions and disinfectant materials [27,28]. Afterward, conventional dental impressions can be disinfected by immersion or spraying techniques.

Disinfection by spraying is considered a suitable method for alginate and polyethers, which are more prone to dimensional distortion after 10 min of immersion, even if the risk of disinfectant inhalation cannot be excluded. On the other hand, the immersion technique, suitable for impression materials less prone to expansion or swelling (i.e., polyvinylsiloxane), grants a full coverage of the disinfectant [27,28].

Glutaraldehyde and chlorine compounds are generally preferable to chemically disinfect dental impressions, with a time of contact established by providers' instructions and specific guidelines [29].

Concerning SARS-CoV-2, some investigations found out that coronavirus contamination can be significantly reduced with 62–71% ethanol or 0.1% sodium hypochlorite solutions within 1 min of exposure; in addition, other biocidal agents (i.e., 70%–75% 2-propanol and 0.23% povidone-iodine) can effectively reduce the viral infectivity [30,31].

Furthermore, the use of automatic mixers is strongly recommended in order to avoid the microbial contamination derived from handling some impression materials (i.e., addition silicones) without gloves.

Regarding the disinfection procedures, the two following physical events are worthy of notice. The first is the negative effect of the disinfection treatment on the dimensional stability of the impression material and the accuracy of its surface detail. The second event is the deactivation effect of the impression material on the disinfecting solution that might decrease the efficiency of the disinfection process [32]. Such events must be kept in check and avoided when possible. For this purpose, it is advisable to follow the instructions provided by the manufacturers of the disinfectants.

Customized trays should be properly disinfected after intraoral try-in and before the use of brushes for impression material adhesives [25].

Regarding digital technologies and optical impressions, a significant advantage is the possibility to autoclave the latest-generation scanners' tips, and proper disinfectant commercial products can be used for other scanner items and scanning devices (i.e., implant-supported scan-bodies) to reduce the risk of cross-contamination, according to manufacturers' suggestions [25,33]. Moreover, intraoral scanner systems provide the advantage of avoiding the transfer of a conventional impression and therefore the risk of contagion to the laboratory via the physical impression or the gypsum cast.

Besides, digital technologies are also useful during the present pandemic for prosthetic planning and the manufacturing of removable or fixed prostheses. Indeed, compared to conventional procedures, the digital workflow allows the fabrication of reliable prostheses in fewer appointments and with quicker chairside times [34].

### 4.2. Prosthodontics Aids

In order to reduce the probability of contagion, maximum attention has to be paid to the disinfection and sterilization of the instruments and aids that can be used during prosthetic clinical practice. For this purpose, it is useful to dwell on the following indications reported in the literature.

The disinfection of some aids, such as occlusal rims or casts, could be achieved by spraying with 5.25% sodium hypochlorite solution and allowing to set for 10 min [35–37]. Moist heat sterilization is performed using a steam autoclave at 121 °C for 15–20 min at 15 lb pressure/square inch, and it is suggested for handpieces, stainless steel instruments, tissue retraction pluggers, dapen dishes, and glass slabs in combination with ethylene oxide at a concentration of 450–800 mg/L [38,39]. Moreover, ethylene oxide might be used for the disinfection of mouth and face mirrors, carbon steel hand instruments, three-way syringes, saliva ejectors, and evacuators [40].

Regarding impression trays, metallic ones should be heat-sterilized via autoclave, chemical vapor, or dry heat and ethylene oxide sterilization. Custom acrylic resin should be disinfected with tuberculocidal hospital disinfectant for reuse during the same patient's next visit, while it is suggested to avoid using plastic trays because they are more difficult to sterilize efficiently [41].

Acrylic dentures must be soaked in glutaraldehyde solution for 12 h after being rinsed with running water and stored in an ultrasonic cleaner. Subsequently, they must be cleaned with running water, scraped with chlorhexidine, and then exposed to chlorine dioxide for 3 min. Then, ethylene oxide is used to sterilize [40]. Regarding dentures made of metal, it

is suggested to spray with 2% glutaraldehyde solution and place the prostheses in a plastic bag for 10 min [32].

### 4.3. Metal–Ceramic Materials

Immersion in glutaraldehydes for the duration indicated by the disinfectant manufacturer can be used to disinfect fixed metal/porcelain prostheses. In addition, fixed prostheses may be disinfected by immersing them in diluted hypochlorite for a brief period of time without causing harm. The higher the noble metal concentration, the lower the risk of harmful effects on the metal core [42]. Care should be taken to limit the amount of time in which metals are exposed to potentially corrosive chemicals. Iodophors might be utilized as well, but there is no evidence to back this up. Fixed metal prostheses can be disinfected with ethylene oxide or even autoclaving if needed. Conversely, unglazed porcelain should not be exposed to any disinfectant because porcelain firing/glazing might be sufficient [42].

Before delivering to the patient, any aid that was treated with a disinfectant must be properly cleaned.

### 4.4. All-Ceramic Materials

All-ceramic materials require surface treatments aimed at favoring the adhesion of the material to the dental surfaces. This surface treatment is called "etching" and mainly concerns the glass ceramics, such as feldspathic or leucite-based ceramics, lithium silicates, and disilicates, in order to cement these materials to the prepared tooth adhesively, increasing the resistance to fracture of the final restoration [43–45]. Regarding zirconia ceramics, they are characterized by the absence of glass in their composition, and the surface treatment takes place through a tribochemical silica coating or low-pressure air abrasion ($Al_2O_3$, 50 μm at 2 bars) for both conventional and adhesive cementations [46–48]. The surface treatment procedures of all-ceramic materials require some rinsing and drying steps, both in the laboratory and in the dental office.

Unfortunately, the rinsing and drying phases lead to nebulization and the formation of bio-aerosols.

Bio-aerosols are liquid/solid particles generated by different sources and are responsible for the transmission of airborne microorganisms, using droplet nuclei (1–5 μm) or droplets (>5 μm). High-speed handpieces, ultrasonic scalers, air turbines, air polishing, and air–water syringes are all sources of bio-aerosols during dental practice. These particles can stay suspended in the air for a variable time and fall, contaminating all the surfaces of dental offices [49]. Tooth preparation procedures or chairside prosthesis modifications require the use of dental handpieces, and the production of aerosols is unavoidable, resulting in a high risk of indirect infections for patients and dental professionals, which is difficult to contain [50]. SARS-CoV-2 spread may occur via respiratory droplets and contact transmission. Hence, it is advisable to meticulously manage the patients to minimize the risk of COVID-19 infections.

In order to avoid the propagation of viruses, various techniques have been proposed to reduce the development of bio-aerosols.

In this regard, the World Health Organization recommends the application of some precautions, as reported in Table 1. The use of personal protective equipment (PPE), such as gloves, long-sleeved uniforms, and eye protection, is mandatory [51–54]. Furthermore, it is important to perform the aerosol-generating procedure in an adequately ventilated room, avoiding the presence of unnecessary individuals and having correct hand and clinic surface hygiene practices [55,56]. A 0.23% povidone-iodine (PVP-I) solution showed virucidal efficacy against SARS-CoV, MERS-CoV, influenzavirus A (H1N1), and rotavirus after 15 s of exposure in vitro; therefore, PVP-I gargle/mouthwash could be useful as a protection measure to reduce the viral load in saliva [57]. In addition, the use of high-speed dental handpieces with special anti-retraction valves would be advisable to

avoid the aspiration of debris and fluids, which could contaminate the tubes within the dental unit [58].

**Table 1.** Guidance for health workers. List of the publications by World Health Organization.

| World Health Organization Guidance for Health Workers | |
|---|---|
| **Publication Title** | **Online Source** |
| **Coronavirus disease (COVID-19) outbreak: rights, roles, and responsibilities of health workers, including key considerations for occupational safety and health** | https://www.who.int/publications-detail/corona-vi-rus-disease-(covid-19)-outbreak-rights-roles-and-responsibilities-of-health-workers-including-key-considerations-for-occupational-safety-and-health (accessed on 21 January 2022). |
| **Infection prevention and control during health care when novel coronavirus (nCoV) infection is suspected** | https://www.who.int/publications-detail/infection-prevention-and-control-during-health-care-when-novel-coronavirus-(ncov)-infection-is-suspected-20200125 (accessed on 21 January 2022). |
| **Health workers exposure risk assessment and management in the context of COVID-19 virus** | https://www.who.int/emergencies/diseases/novel-coronavirus-2019/technical-guidance/health-workers (accessed on 21 January 2022). |
| **Rational use of personal protective equipment for coronavirus disease (COVID-19)** | https://www.who.int/emergencies/diseases/nov-el-coronavirus-2019/technical-guidance/health-workers (accessed on 21 January 2022). |
| **Advice on the Use of Masks** | https://www.who.int/publications-detail/advice-on-the-use-of-masks-in-the-community-during-home-care-and-in-healthcare-settings-in-the-context-of-the-novel-coronavirus-(2019-ncov)-outbreak (accessed on 21 January 2022). |
| **Home care for patients with suspected novel coronavirus (nCoV) infection presenting with mild symptoms and management of contacts** | https://www.who.int/publications-detail/home-care-for-patients-with-suspected-novel-coronavirus-(ncov)-infection-presenting-with-mild-symptoms-and-management-of-contacts (accessed on 21 January 2022). |
| **Q&A on infection prevention and control for health care workers caring for patients with suspected or confirmed 2019-nCoV** | https://www.who.int/news-room/q-a-detail/q-a-on-infection-prevention-and-control-for-health-care-workers-caring-for-patients-with-suspected-or-confirmed-2019-ncov (accessed on 21 January 2022). |
| **Water, sanitation, hygiene and waste management for COVID-19** | https://www.who.int/publications-detail/water-sanitation-hygiene-and-waste-management-for-covid-19 (accessed on 21 January 2022). |
| **Guide to local production of WHO-recommended Handrub Formulations** | https://www.who.int/emergencies/diseases/novel-coronavirus-2019/technical-guidance/health-workers (accessed on 21 January 2022). |
| **IPC guidance for long-term care facilities in the context of COVID-19** | https://www.who.int/emergencies/diseases/novel-coronavirus-2019/technical-guidance/health-workers (accessed on 21 January 2022). |

The publications reported in Table 1 are available at https://www.who.int/emergencies/diseases/novel-coronavirus-2019/technical-guidance/health-workers (accessed on 21 January 2022).

Regarding all-ceramic materials, different disinfectant methods might be used, such as ultrasonication in a bath containing 4% Lysetol AF for 5 min at room temperature [59], pellets soaked in 75% alcohol applied on the crown surfaces [60], immersion in 2% glutaraldehyde for 30 min [61], or in 2% glutaraldehyde for 10 h [61]. Unfortunately, the scientific literature does not define the best disinfection method, and its effectiveness against SARS-CoV-2 is not clear. Nevertheless, it is advisable to follow a sterilization process in an autoclave, although an ideal sterilization protocol for all-ceramic or metal–ceramic materials has not yet been defined. Porto et al. [61] reported a protocol where an autoclave sterilization process was performed at the temperature of 127 °C for 15 min, and at a pressure of 1.5 Kgf/cm$^2$.

### 4.5. Pre-Check Triages

Considering all the risks of contagion from SARS-CoV-2 and the listed methods to deal with them, it is suggested to follow a process of pre-check triages via telephone for the evaluation of patients, as proposed in Table 2 [21,55,62,63].

**Table 2.** Pre-check telephonic triage for COVID-19 management in dental practice.

| Telephonic triage |
| --- |
| Date:<br>Name:<br>Surname:<br>Reason for the visit: |
| Have you had at least one of the following symptoms of respiratory illness in the last 14 days?<br>• fever (T > 37.5 °C);<br>• cough;<br>• sore throat;<br>• rhinorrhea;<br>• dyspnea;<br>• asthenia;<br>• dysgeusia or hyposmia. |
| In the past 14 days, have you or your family member had:<br>• national or international travel;<br>• contact with a suspected or confirmed case of COVID-19;<br>• attended or worked in healthcare facilities. |
| Did you take the test for SARS-CoV-2 and have an uncertain or positive result? |

First of all, the patient's personal data must be collected, such as the name and surname with the date and reason for the visit. Subsequently, telephone questions should be asked to assess if the patient had had at least one of the following symptoms of respiratory illness, in the last 14 days: fever (T > 37.5 °C), cough, sore throat, rhinorrhea, dyspnea, asthenia, dysgeusia, or hyposmia. If there is a history of at least one of them, then the clinical team will need to assess whether they may be related to SARS-CoV-2 or other pathologies.

The patient should be asked if, in the past 14 days, he/she or his/her family member had national or international travel, had contact with a suspected or confirmed case of COVID-19, or attended or worked in healthcare facilities. Finally, it is be necessary to ask if the patient has taken the test for SARS-CoV-2 and received an uncertain or positive result (Table 2).

To sum up, the dental staff has to ask patients about recent travels, contacts with confirmed or suspected cases of COVID-19, and if they present symptoms of respiratory disease (i.e., fever or cough). In cases of positive responses, the clinician appointment should be postponed for 14 days, and patients need to be advised to contact their primary care physician [21,55].

### 4.6. Limitations of the Search Methodology

The present paper is not a systematic review or meta-analysis. The chosen search methodology offers a summary of the current literature and sheds light on a general problem rather than a synthesized finding or response to a particular question.

With this review, the presence of statements about the formal synthesis and the quality of evidence is excluded as well as a critical assessment of the risk of bias.

Therefore, the present paper does not provide any statistically proven findings, such as a systematic review or meta-analysis. Conversely, this review qualitatively summarized evidence, providing an overview on the risk management of COVID-19 infection in dental practice, in particular focusing on prosthodontics and all-ceramic materials.

Furthermore, the bibliographic search included a limited number of records since the literature regarding COVID-19, although growing, is quite scant. Further in vitro and in vivo studies and, in particular, randomized clinical trials (RCTs) are needed to validate the outcomes of this paper.

*4.7. Future Perspective*

It is not easy to predict what may happen in the future regarding the COVID-19 pandemic, as two factors do not allow us to let our guard down. First of all, vaccination coverage is not yet complete, and there are many unvaccinated people around the world. Second, the emergence of new variants, such as Omicron, continues. Therefore, in the future, it may be helpful to have dental staff subjected to both serological screening and immunization. This could be critical to preserving the health of both healthcare professionals and patients [63,64].

*4.8. Summary of Risk Management of COVID-19*

As reported in the present literature search, it is essential to follow dedicated decontamination and disinfection protocols for conventional impressions. Conversely, the use of intraoral scanners allows the advantage of avoiding the transport of a conventional impression and therefore the risk of contagion to the laboratory through the physical impression or the gypsum cast.

Furthermore, a digital workflow should be adopted because it can reduce the number and duration of appointments during prosthesis manufacturing.

In any case, a conventional disinfection and decontamination protocol of any aid used in clinical practice must be followed. Moreover, particular attention should be paid to the bio-aerosols that can be generated during some procedures, such as the preparation of dental abutments, chairside prosthesis modifications, or the rinsing and drying phases that characterize the adhesion of all-ceramic restorations to the dental surface. In addition to the use of personal protective equipment, it is strongly recommended to work in a ventilated room, avoiding the presence of unnecessary individuals and using handpieces with special anti-retraction valves.

Finally, a protocol of pre-check triages via telephone should be adopted in order to further reduce the potential risk of contagion in dental offices.

**5. Conclusions**

In the medical field, dental offices are undoubtedly at high risk of spreading SARS-CoV-2 infection due to face-to-face contact with patients and continuous exposure to saliva during dental procedures. Universal precautions have been issued by the WHO to control the spread of infection, protecting both patients and health workers. Among health workers, dental practitioners (i.e., dentists, dental hygienists, dental nurses, and dental technicians) face the greatest risk of COVID-19 infection. Hence, meticulous guidelines have to be followed to minimize the contamination. In this period of a global health crisis, dentists are recommended to defer elective dental treatments, including prosthodontic ones, focusing only on emergency care.

Nonetheless, this novel SARS-CoV-2 could persist over time, and asymptomatic patients may represent a source for virus re-emergence. This implies that the clinicians should behave cautiously to reduce the potential risk of SARS-CoV-2 cross-infection, following meticulous contamination control procedures, mainly during high-risk practices, such as prosthodontics.

In this regard, it is important to know the stages in which contagion can occur during daily clinical practice and how to avoid the risk of infection.

**Author Contributions:** Conceptualization, R.S. and F.Z.; methodology, G.R. and P.B.; validation, R.L. and M.I.D.M.; formal analysis, M.B.; investigation, G.R. and P.B.; resources, R.S. and F.Z.; data curation, R.L. and M.I.D.M.; writing—original draft preparation, M.I.D.M. and G.R.; writing—review

and editing, R.L.; visualization, F.Z.; supervision, R.S.; project administration, R.L. All authors have read and agreed to the published version of the manuscript.

**Funding:** This research received no external funding.

**Institutional Review Board Statement:** Not applicable.

**Informed Consent Statement:** Not applicable.

**Data Availability Statement:** The datasets used and/or analyzed during the current study are available from the corresponding author on reasonable request.

**Conflicts of Interest:** The authors declare no conflict of interest.

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
