# Peer review of "A Review on Risk Management of Coronavirus Disease 19 (COVID-19) Infection in Dental Practice: Focus on Prosthodontics and All-Ceramic Materials"

_prosthesis, doi:10.3390/prosthesis4030028_

Round 1
Reviewer 1 Report
This study aimed to present an overview to limit the risk of transmission of COVID-19 infections during prosthetic procedures in dental offices. The concepts discussed and the information provided in this article are too general and without further explanation. This information is already provided by the regulatory authorities to dentists with more specificity like https://www.ada.org.au/Dental-Professionals/Publications/Infection-Control/Guidelines-for-Infection-Control/Guidelines-for-Infection-Control-V4.aspx. In general, this article is not useful for the clinicians since they are already following standard precaution for their practice, and this research is not providing enough data about transmission precautions for infection control.
Author Response
Point 1:This study aimed to present an overview to limit the risk of transmission of COVID-19 infections during prosthetic procedures in dental offices. The concepts discussed and the information provided in this article are too general and without further explanation. This information is already provided by the regulatory authorities to dentists with more specificity like https://www.ada.org.au/Dental-Professionals/Publications/Infection-Control/Guidelines-for-Infection-Control/Guidelines-for-Infection-Control-V4.aspx. In general, this article is not useful for the clinicians since they are already following standard precaution for their practice, and this research is not providing enough data about transmission precautions for infection control.
Response 1. The authors appreciate the observation of Reviewer #1. However, while keeping in mind the international guidelines, the authors would like to point out that they have focused in detail on prosthetic aspects as demonstrated in the dedicated paragraphs on prosthodontics aids (paragraph 4.2), on metal-ceramic products (paragraph 4.3) and on all-ceramics ones (paragraph 4.4).
Reviewer 2 Report
In this manuscript authors review on risk management of Coronavirus Disease 19 (COVID-19) infection in dental practice: focusing on Prosthodontics and all-ceramic materials. Further suggestions need to address by the authors before it publishes:
1. Introduction: Authors need to improve the introduction part. Please cite some recent literature to enrich the introduction part, especially COVID 19
2. Line 316: Bio-aerosols are liquid/solid particles generated by different sources, kindly mentioned the name of the sources.
3. Kindly add more figures from recent literatures to improve the quality of the manuscript.
4. Authors need to add future perspective in this manuscript.
5. The authors should summarize the current approaches for thr management of Coronavirus Disease 19 and compare their advantages and disadvantages in order to draw the reader's attention
Author Response
The authors thank Reviewer #2 for the valuable and constructive suggestions to improve the manuscript. A point-by-point response is included below. The authors’ responses are in bold. Changes suggested by reviewers are marked up using the “Track Changes” function in the manuscript, as requested by the Editorial Office.
Point 1: Introduction: Authors need to improve the introduction part. Please cite some recent literature to enrich the introduction part, especially COVID 19.
Response 1. The authors are grateful to Reviewer #2, for the valuable comment that enhances the quality of the manuscript. The requested modification was made to the Introduction section.
Point 2: Line 316: Bio-aerosols are liquid/solid particles generated by different sources, kindly mentioned the name of the sources
Response 2: The authors thank Reviewer #2 for his/her observation and the different sources of bio-aerosols were clarified in the text (page 10, lines 343-344).
Point 3: Kindly add more figures from recent literatures to improve the quality of the manuscript.
Response 3. The requested modification was made, and more figures were included in the main manuscript.
Point 4: Authors need to add future perspective in this manuscript.
Response 4. According to the Reviewer’s comment, the paragraph “4.7 Future perspective” was added to the manuscript.
Point 5: The authors should summarize the current approaches for thr management of Coronavirus Disease 19 and compare their advantages and disadvantages in order to draw the reader's attention.
Response 5. The requested modification was made to the manuscript and the following paragraph was added according to the Reviewer’s indication “4.8 Summary of risk management of COVID-19”
Reviewer 3 Report
Dear Editor,
the authors have submitted an interesting review about the risk management of COVID-19 in Prosthodontics: the manuscript is updated and coherent. A general revision of English writing is suggested, as minor typos were highlighted thorough the text. A few points were raised after reading the paper and the following explanations are necessary:
- As regards the calibration process, did the internal reviewers present the same level of experience?
- How was the number of 30 references and 12 full-text stated for calibration screening?
- As to optical impressions and IOS systems, are all scanner tips autoclavable?
- The Conclusions paragraph should be summarized and has not to repeat statements previously reported in the Introduction and/or Discussion sections.
According to the above mentioned points, I suggest to accept the manuscript for publication after minor revisions.
Author Response
The authors thank Reviewer #3 for the appreciation of the paper. Moreover, the authors are grateful for the valuable and constructive suggestions to improve the manuscript. A point-by-point response is included below. The authors’ responses are in bold. Changes suggested by reviewers are marked up using the “Track Changes” function in the manuscript, as requested by the Editorial Office.
Point 1: As regards the calibration process, did the internal reviewers present the same level of experience?
Response 1: The internal reviewers present the same level of experience and it has been clarified in the manuscript.
Point 2: How was the number of 30 references and 12 full-text stated for calibration screening?
Response 2: The number of 30 references and 12 full-text was based on both convenience criteria and literature. This size is not relevant to the statistical results and outcome of the present study, as it is useful only for an exercise among the reviewers. Moreover, in the literature, some authors used a smaller sample size (n=10) than in our study (n=12) for the full-text screening process [Takajashi et al 2014].
-Glover Takahashi, Susan et al. “The epidemiology of competence: protocol for a scoping review.” BMJ open vol. 4,12 e006129. 31 Dec. 2014, doi:10.1136/bmjopen-2014-006129
Point 3: As to optical impressions and IOS systems, are all scanner tips autoclavable?
Response 3: The authors confirm that the latest generation scanners all have heat-sterilizable tips and it has been clarified in the manuscript.
Point 4: The Conclusions paragraph should be summarized and has not to repeat statements previously reported in the Introduction and/or Discussion sections.
Response 4: The Conclusion paragraph was rewritten following the Reviewer’s comment,
According to the above mentioned points, I suggest to accept the manuscript for publication after minor revisions.
The authors are grateful to Reviewer #3 for his/her appreciation of the manuscript.
Round 2
Reviewer 1 Report
The authors addressed all my concerns and the manuscript is significantly improved.